# Independent *COL5A1* Variants in Cats with Ehlers-Danlos Syndrome

**DOI:** 10.3390/genes13050797

**Published:** 2022-04-29

**Authors:** Sarah Kiener, Neoklis Apostolopoulos, Jennifer Schissler, Pascal-Kolja Hass, Fabienne Leuthard, Vidhya Jagannathan, Carole Schuppisser, Sara Soto, Monika Welle, Ursula Mayer, Tosso Leeb, Nina M. Fischer, Sabine Kaessmeyer

**Affiliations:** 1Institute of Genetics, Vetsuisse Faculty, University of Bern, 3001 Bern, Switzerland; sarah.kiener@vetsuisse.unibe.ch (S.K.); fabileuthard@gmail.com (F.L.); vidhya.jagannathan@vetsuisse.unibe.ch (V.J.); 2Dermfocus, University of Bern, 3001 Bern, Switzerland; sara.soto@vetsuisse.unibe.ch (S.S.); monika.welle@vetsuisse.unibe.ch (M.W.); sabine.kaessmeyer@vetsuisse.unibe.ch (S.K.); 3Anicura Kleintierspezialisten Augsburg, 86157 Augsburg, Germany; apostolopou2@wisc.edu (N.A.); info@haut-tier-arzt.de (U.M.); 4Department of Medical Sciences, School of Veterinary Medicine, University of Wisconsin, 2015 Linden Drive, Madison, WI 53706, USA; 5Department of Clinical Sciences, James L. Voss Veterinary Teaching Hospital, Fort Collins, CO 80525, USA; jennschiss@att.net; 6Division of Veterinary Anatomy, Vetsuisse Faculty, University of Bern, 3001 Bern, Switzerland; pascal-kolja.hass@fu-berlin.de; 7VET Zentrum AG, 8422 Pfungen, Switzerland; cschuppisser@vetzentrum.ch; 8Institute of Animal Pathology, Vetsuisse Faculty, University of Bern, 3001 Bern, Switzerland; 9Dermatology Unit, Clinic for Small Animal Internal Medicine, Vetsuisse Faculty, University of Zurich, 8057 Zurich, Switzerland; nfischer@vetclinics.uzh.ch

**Keywords:** *Felis catus*, dermatology, genodermatosis, skin, precision medicine, animal model

## Abstract

We investigated four cats with similar clinical skin-related signs strongly suggestive of Ehlers-Danlos syndrome (EDS). Cases no. 1 and 4 were unrelated and the remaining two cases, no. 2 and 3, were reportedly siblings. Histopathological changes were characterized by severely altered dermal collagen fibers. Transmission electron microscopy in one case demonstrated abnormalities in the collagen fibril organization and structure. The genomes of the two unrelated affected cats and one of the affected siblings were sequenced and individually compared to 54 feline control genomes. We searched for private protein changing variants in known human EDS candidate genes and identified three independent heterozygous *COL5A1* variants. *COL5A1* is a well-characterized candidate gene for classical EDS. It encodes the proα1 chain of type V collagen, which is needed for correct collagen fibril formation and the integrity of the skin. The identified variants in *COL5A1* are c.112_118+15del or r.spl?, c.3514A>T or p.(Lys1172*), and c.3066del or p.(Gly1023Valfs*50) for cases no. 1, 2&3, and 4, respectively. They presumably all lead to nonsense-mediated mRNA decay, which results in haploinsufficiency of *COL5A1* and causes the alterations of the connective tissue. The whole genome sequencing approach used in this study enables a refinement of the diagnosis for the affected cats as classical EDS. It further illustrates the potential of such experiments as a precision medicine approach in animals with inherited diseases.

## 1. Introduction

Ehlers-Danlos syndromes (EDS) are heritable disorders of connective tissue that share several clinical features, such as skin hyperextensibility, joint hypermobility, abnormal wound healing, and easy bruising [1]. Based on clinical findings and the mode of inheritance, different subtypes are recognized in humans, caused by alterations in 20 different genes [1,2,3] (Appendix A). One subtype is classical EDS, which is caused by defects in type V collagen and results from pathogenic variants in *COL5A1* or *COL5A2* [2,4,5,6,7,8,9]. Type V collagen is a major regulator of collagen fibril diameter and has a critical role during the early process of collagen fibril nucleation [1]. Targeted and spontaneous animal models for EDS have recently been reviewed [10].

Spontaneous cases of EDS have been previously described in cats as cutaneous asthenia or dermatosparaxis (OMIA 000327-9685; [10,11,12,13,14,15,16,17,18,19,20,21,22,23]). A candidate causative variant in *COL5A1* has been identified in one cat. The affected cat carried a heterozygous single base pair deletion, c.3420del, resulting in a frameshift and premature stop codon [24].

In this study, we characterized the clinical and histopathological phenotype of four cats with clinical signs of classical EDS and performed genetic analyses to search for the causative genetic variants.

## 2. Materials and Methods

### 2.1. Animal Selection

This study included four EDS-affected cats, one Bengal cat (case no. 1), two presumably related shelter cats reported to be Bombay with unknown lineage (cases no. 2 and 3), and one domestic shorthair cat (case no. 4). For case no. 4, we additionally had samples of two unaffected siblings and both parents. Samples from 54 additional genetically diverse control cats of the Vetsuisse Biobank were used for the whole genome sequencing data analysis (Appendix A).

### 2.2. Clinical and Histopathological Examinations

All cats underwent clinical examination. For case no. 4, hematological and biochemical parameters were measured, and ELISA tests for Feline Leukemia Virus (antigens), Feline Immuno-deficiency Virus (antibodies), and Feline Coronavirus (antibodies) were performed (IDEXX Laboratories, Kornwestheim, Germany). Furthermore, a skin intradermal allergy test (Artuvetrin Haut Test, Nextmune, Leipzig, Germany) and a serum allergy test (IDEXX Laboratories, Kornwestheim, Germany) were carried out.

Skin biopsies were obtained from cases no. 1, 2, and 4. The biopsies were fixed in 10% buffered formalin and processed routinely. The slides were stained with hematoxylin and eosin (H&E), periodic acid–Schiff (PAS), and Masson’s Trichrome (MT), prior to histopathological examination.

### 2.3. Transmission Electron Microscopy (TEM)

Skin punch biopsies from case no. 4 and the skin punch biopsy of an age-matched control cat were fixed in a 2.5% glutaraldehyde/0.1 M cacodylate buffer solution, then washed in 0.1 M cacodylate buffer (cacodylic acid sodium salt trihydrate; Merck KGaA, Darmstadt, Germany), and afterwards contrasted for 4 h in 1% osmium tetroxide (Polysciences Europe GmbH, Hirschberg, Germany). Samples were then dehydrated in an ascending series of ethanol and embedded in a mixture of Epon 812 (epoxy resin), dodecenylsuccinic anhydride (plasticizer), methylnadic anhydride (hardener), DMP 30 (catalyst) (all: Merck KGaA, Darmstadt, Germany), and polymerized at 60 °C for 5 days. Semi- and ultrathin sections were cut with an ultra-microtome Reichert Ultracut S (Leica, Wetzlar, Germany). Semithin sections (0.5 µm) were stained with 1% Toluidine blue solution (Merck KGaA, Darmstadt, Germany). From the semi-thin sections, representative areas for ultrastructural analysis were selected by light microscopy (Axioimager, Zeiss, Oberkochen, Germany). Ultrathin (80 nm) sections were mounted on copper-grids (Agar Scientific Ltd., Stansted, Essex, UK), contrasted with lead citrate and UranyLess (Electron Microscopy Sciences, Hatfield, PA, USA), and examined with a transmission electron microscope (CM12; Zeiss, Oberkochen, Germany).

### 2.4. DNA Extraction

Genomic DNA was isolated from EDTA blood using the Maxwell^®^ RSC Whole Blood DNA Kit with the Maxwell^®^ RSC instrument (Promega, Dübendorf, Switzerland).

### 2.5. Whole-Genome Sequencing and Variant Calling

Illumina TruSeq PCR-free DNA libraries were prepared for cases no. 1, 2, and 4 and sequenced on a NovaSeq 6000 instrument. The sequence data were submitted to the European Nucleotide Archive with the study accession PRJEB7401 and sample accessions SAMEA5885921 (case no. 1), SAMEA5885927 (case no. 2), and SAMEA7853381 (case no. 4). Mapping and alignment were performed as described [25]. Variant calling was performed using GATK HaplotypeCaller [26] in gVCF mode as described [25]. To predict the functional effects of the called variants, SnpEff version 4.3t software [27] together with NCBI annotation release 104 for the felCat9.0 genome reference assembly was used. For variant filtering, we used 54 control genomes (Appendix A).

### 2.6. Gene Analysis

We used the felCat9.0 reference genome assembly and NCBI annotation release 104. Numbering within the feline *COL5A1* gene corresponds to the NCBI RefSeq accession numbers XM_023242950.1 (mRNA) and XP_023098718.1 (protein).

### 2.7. PCR and Sanger Sequencing

Candidate variants were confirmed and genotyped either by direct Sanger sequencing of PCR amplicons or fragment length analysis of PCR products on a 5200 Fragment Analyzer (Agilent, Basel, Switzerland). PCR products were amplified from genomic DNA using AmpliTaq Gold 360 Mastermix (Thermo Fisher Scientific, Reinach, Switzerland) together with a forward and reverse primer (Appendix A). After treatment with exonuclease I and alkaline phosphatase, amplicons were sequenced on an ABI 3730 DNA Analyzer (Thermo Fisher Scientific). Sanger sequences were analyzed using the Sequencher 5.1 software (GeneCodes, Ann Arbor, MI, USA).

## 3. Results

### 3.1. Clinical History and Examination

Case no. 1 was a 1-year-old purebred Bengal cat with a history of multiple repetitive skin lesions after minimal trauma occurring since the age of 6 months. At the time of presentation, a scarring alopecic lesion of about 5 × 5 cm was visible on the dorsal neck, which resulted from a previous trauma. Furthermore, the skin was diffusely thinner and much more elastic compared to the skin of a healthy young Bengal cat. In addition, the patient presented with multiple small skin tears distributed over the entire trunk/back area (Figure 1). The cat had no pruritus and there were no signs of active inflammation. Skin cytology, skin scrapings, and the trichogram were unremarkable.

Cases no. 2 and 3, presumably related female spayed Bombay cats, were presented for multiple wounds in the hyperextensible skin predominantly at the neck, base of ear, and shoulders (Figure 2a). The wounds occurred after self-trauma or exuberant play. Case no. 2 additionally showed spasms of the cutaneous trunci followed by frantic self-mutilation since adoption at 8 weeks of age. At 11 months of age, case no. 2 also experienced multiple partial and grand mal seizures necessitating phenobarbital therapy. Case no. 3 experienced a grade 3 patellar luxation at 18 months of age presumed to be related to joint hypermobility due to EDS. Excessive facial skin folds were apparent (Figure 2b).

Case no. 4 was a 2-year-old male castrated indoor DSH cat that was examined for recurrent laceration wounds on the neck and shoulders since adoption (one year before), and moderate head and neck pruritus (observed all year-round with exacerbation in June and August). Hematology and biochemistry (including basal cortisol) were normal, and Feline Leukemia Virus, Feline Immunodeficiency Virus, and Feline Coronavirus were negative one year before referral. On physical examination, a 1 cm scar on the ventral neck and a 0.5 cm crust above the right eye were noted. The skin extensibility index was 22% (Figure 3a). No joint laxity, hernias, ocular, or heart abnormalities were found. Neither ecto- and endoparasite treatment of both cats, nor an eight-week elimination diet improved the pruritus. Over the course of the disease, the cat developed new lesions due to pruritus when treated only with oral antihistamines (Fenistil, GlaxoSmithKline Consumer Healthcare GmbH & Co. KG, Munich, Germany) (Figure 3b,c). With short courses of systemic prednisolone (injections of 2 mg/kg, or orally 0.6 mg/kg once daily, as needed), the scratching stopped, and the lesions healed. Systemic medication was withdrawn as indicated prior to biopsy sampling. A welfare score (8; 0–21) was not suggestive of a psychogenic induced pruritus [28]. Based on all the above and since the patient fulfilled Favrot’s criteria [29], which help to establish the diagnosis of atopic dermatitis, a feline atopic skin syndrome (FASS) was diagnosed and a concomitant collagen disorder leading to lacerations due to pruritus (traumatic splitting) was suspected [30]. A skin intradermal allergy test as well as a serum allergy test were negative and could not identify the triggering agents of FASS. The owner declined antiallergic therapies other than antihistamines, and the cat was euthanized 2.5 years after the first presentation due to a very large laceration wound and deterioration, despite surgical and emergency treatment.

Two unaffected siblings and both parents of the patient were also clinically evaluated. They did not have any history compatible with allergy and no abnormality was noted. Their skin extensibility indices were normal (father: 15.6%, mother: 6.6%, brother 17.6%, sister: 10.7%).

### 3.2. Histopathological Examination

The pathological changes in the skin biopsies were characterized by severely altered dermal collagen fibers and a slightly thinner epidermis. Changes were more severe in the superficial and mid dermis but were also present in the deep dermis. The abnormal collagen fibers were haphazardly arranged, shortened, and sometimes curled. They were uneven in length and width. Many fibers were very wispy and the interfibrillar spaces were widened. Multifocally small areas of hemorrhage were seen (Figure 4).

### 3.3. Ultrastructural Examination

TEM analysis of case no. 4 skin connective tissue revealed abnormalities in the collagen fibril organization and structure. Abnormal findings from the reticular part of the dermis were loosely packed and disorganized collagen fibers (Figure 5b,d). Individual collagen fibers contained curled fibrils (Figure 5b,d). Higher magnifications showed cross-sections of collagen fibrils with different diameters and irregularly outlined fibrils within a fiber (Figure 5f,h). In contrast, the connective tissue of an age-matched control cat contained densely packed collagen fibers with parallel-aligned fibril organization (Figure 5a,c). Cross-sections of fibrils showed uniform diameters and regularly shaped, almost round outlines (Figure 5e,g).

### 3.4. Genetic Analysis

To identify the causative genetic variants, the genomes of three affected cats (cases no. 1, 2, and 4) were sequenced. Case no. 3 was presumably related to case no. 2 and assumed to share the same pathogenic variant. Since the other cats were not known to be related, we hypothesized that their phenotypes were due to independent pathogenic variants. We therefore performed an individual variant filtering in each of the affected cats by comparing them to the genomes of 54 control cats. We focused our search on private protein-changing variants in the 20 known EDS candidate genes (Table 1 and Appendix A).

In case no. 1, we identified one heterozygous private protein-changing variant within an EDS candidate gene (Figure 6a). The variant, *COL5A1*:c.112_118+15del, is a 22 bp deletion that removes the exon1/intron1 boundary (Table 2, Figure 6b). The presence of the variant was confirmed by PCR and fragment length analysis.

In case no. 2, we identified another single candidate variant, a heterozygous nonsense variant in *COL5A1*, predicted to truncate 36% of the coding sequence (Table 2). We confirmed identical heterozygous genotypes in cases no. 2 and 3 by Sanger sequencing (Figure 6a,c).

In case no. 4, we identified three heterozygous variants in known EDS candidate genes, a frameshift deletion in *COL5A1* and two missense variants in *TNXB*. The targeted genotyping of all three variants in the complete family comprising both parents and two additional unaffected littermates excluded the *TNXB* variants (Appendix A). The remaining variant, *COL5A1*:c.3066del, is a one base pair deletion, resulting in a frameshift and predicted to truncate 44% of the open reading frame (Table 2, Figure 6a,d). The genotyping results in the family further revealed that *COL5A1*:c.3066del represented a de novo variant as the mutant allele was absent from the leukocyte DNA of both parents.

## 4. Discussion

In this study, we describe four cats with classical EDS. All cats were characterized by similar clinical skin-related symptoms strongly suggestive of EDS [2]. Histopathology of the examined skin biopsies was also suggestive of EDS. In addition, TEM on case no. 4 identified abnormal morphological features indicative of EDS, such as variable diameters of collagen fibrils, disorganized, distorted collagen fibers, and irregularly outlined fibrils, which have been reported in the literature to be present in most patients with a genetically confirmed diagnosis of EDS [31,32,33].

Whole genome sequencing was performed to confirm the suspected clinical and histological diagnosis and to identify the underlying causative genetic variants in all four cats. We identified three independent heterozygous private protein-changing variants in *COL5A1* in the four investigated affected cats. A de novo mutation event was experimentally confirmed for case no. 4. We assume that the pathogenic *COL5A1* variants in the other affected cats were also due to recent de novo mutation events. For cases no. 2 and 3, the most likely scenario was a germline mutation in one of their parents. These results resemble previously described findings in a cat and three dogs affected by EDS due to de novo mutations in *COL5A1* [24,34].

*COL5A1* encodes the proα1 chain of type V collagen, which together with type I collagen forms heterotypical fibrils. Type V collagen is a major regulator of collagen fibrillogenesis, and the reduction in collagen V expression results in fewer collagen I fibrils with increased diameters [35,36]. Correct fibril formation is required for the integrity of the skin [1].

In humans, the functional loss of one *COL5A1* allele is the most commonly reported molecular mechanism in classical EDS [36,37]. As type V procollagen molecules cannot accommodate more than a single proα2(V) chain, the reduction in available proα1(V) chains results in the production of about half the normal amount of type V collagen [38]. We assume that all three newly reported feline variants lead to nonsense-mediated mRNA decay, which results in a haploinsufficiency of *COL5A1* and causes the alterations of the connective tissue.

Case no. 2 experienced epilepsy and presumptive hyperesthesia signs. Notably, an EDS-affected Burmese cat was described as experiencing prolonged episodes of disturbing vocalizations that were considered to be partial seizures. Upon necropsy, this cat was found to have flattened cerebral gyri [14]. Epilepsy has been noted in human patients with EDS and is associated with many structural brain defects, including periventricular heterotopia in patients with *COL5A1* variants [39,40]. EDS-associated small fiber polyneuropathy in human patients manifests as acute, unpredictable burning pain, muscle cramping, and also compulsive scratching due to severe episodic itch [41]. Given comparable observations in human EDS patients and the earlier reported EDS-affected Burmese cat, it is reasonable to suspect that epilepsy and hyperesthesia were likely part of the EDS phenotype in case no 2.

Case no. 4 was diagnosed with EDS and concurrent FASS. Diagnosis of FASS can be difficult since it is a clinical diagnosis, and no diagnostic tests exist to diagnose or exclude FASS. Allergy tests such as serology and intradermal testing only help to identify the triggering agents to perform allergen specific immunotherapy. However, FASS cases have sometimes been described to have negative allergy testing results [30]. Furthermore, the cat was presented with head and neck pruritus. This is a skin reaction pattern that is consistent with the diagnosis of FASS after excluding other possible causes, as performed in our case [30]. Therefore, based on the clinical presentation and the fact that the cat showed a seasonal exacerbation, as well as the response to histamines (partial) and steroids (complete), we concluded that the itchiness has an underlying allergic cause. A neurological component of the itch therefore seemed unlikely, but it could not be completely excluded without further diagnostic imaging or necropsy, which was declined by the owner.

EDS-affected pets with other concurrent diseases inducing self-trauma (such as itchiness in case no. 4 due to FASS) have exacerbated symptoms that might lead to euthanasia. A faster whole genome sequencing-based diagnosis of the primary collagen genetic disease might lead to a better overall management of the patients, with the potential of decreasing the euthanasia rates of such complicated cases. A precise classification of EDS relies on molecular confirmation with identification of a causative variant [2]. This study illustrates the potential of whole genome sequencing as a precision medicine approach in animals with inherited diseases.

Consensus criteria for the interpretation of genetic variants in domestic animals have not yet been defined. However, such criteria have been established for human patients and they have proven highly useful to provide a standardized classification reflecting the strength of evidence for pathogenicity [42]. When we apply the human guidelines to the investigated cats, all three reported *COL5A1* variants would be classified as pathogenic according to the ACMG/AMP criteria [42]. All three reported variants represent null variants (very strong evidence of pathogenicity, PVS1), are absent from controls (moderate evidence of pathogenicity, PM2), and the patients’ phenotypes are highly specific for a disease with a single genetic etiology (supporting evidence of pathogenicity, PP4). For case no. 4 we have the confirmed de novo mutation event and family segregation data as additional criteria for pathogenicity.

## 5. Conclusions

We characterized four cats with EDS and identified three independent causal variants in the *COL5A1* gene. De novo mutation events leading to haploinsufficiency of *COL5A1* appear to represent an important molecular etiology in feline EDS.

## Figures and Tables

**Figure 1 genes-13-00797-f001:**
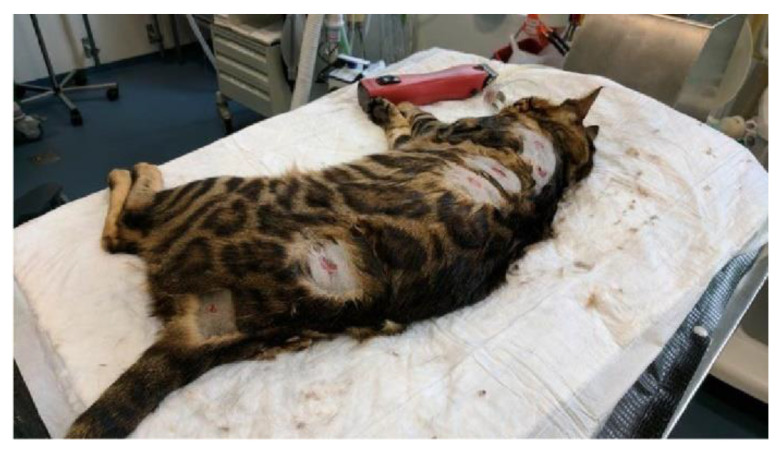
Clinical image of the EDS-affected cat no. 1. Multiple small skin tears were distributed over the entire back.

**Figure 2 genes-13-00797-f002:**
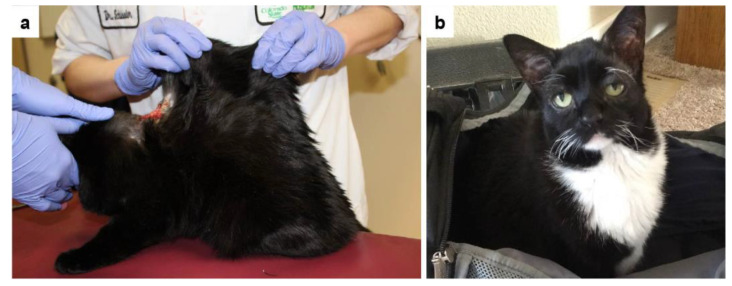
Clinical images of the EDS-affected cats no. 2 and 3. (**a**) Severe hyperextensibility of the dorsal skin. The skin extensibility index was 19% [20]. Note an extensive laceration on the back. (**b**) Excessive skin tissue is visible on the face of case no. 3.

**Figure 3 genes-13-00797-f003:**
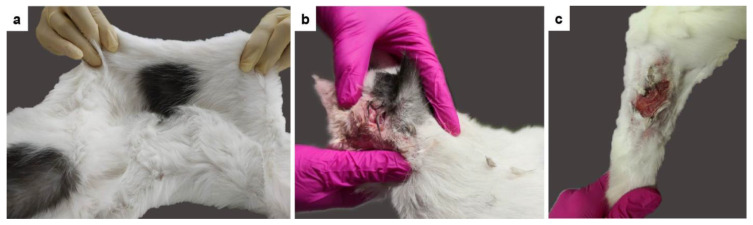
Clinical images of the EDS-affected cat no. 4. (**a**) The abnormal extensibility of the dorsal skin is clearly visible. The skin extensibility index was 22%. (**b**,**c**) Lacerations of the skin on the head and the leg.

**Figure 4 genes-13-00797-f004:**
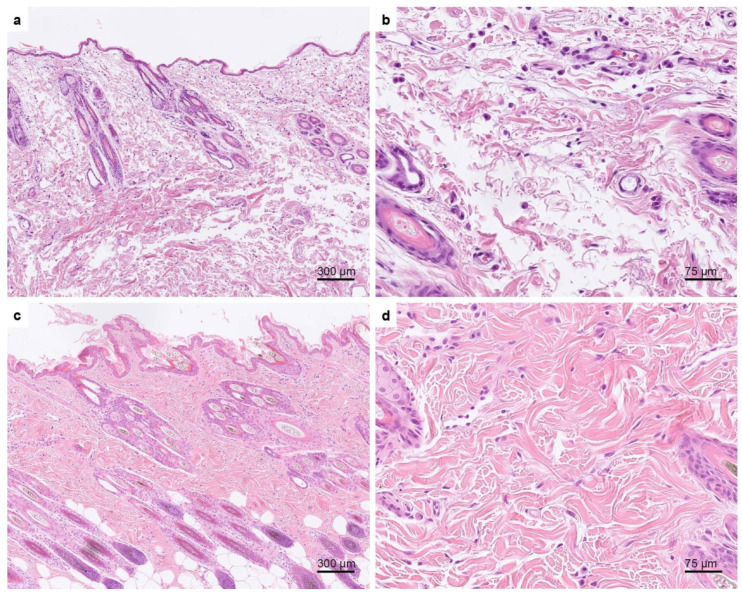
Hematoxylin and eosin-stained skin biopsies of case no. 4 and an age-matched control. (**a**,**b**) Histopathological changes seen in the affected cat. The epidermis appears thinner than in the skin biopsies of the age-matched control cat (**c**,**d**). (**a**) The dermal collagen fibers are haphazardly arranged, shortened, and sometimes curled. They were uneven in length and width. Many fibers are very wispy and the interfibrillar spaces are widened. (**b**) Higher magnification of the dermal changes seen in the affected cat. (**c**) Normal skin of an age-matched control cat. The dermal collagen fibers are much thicker and longer and the interfibrillar spaces are smaller than in the affected cat. (**d**) Higher magnification of normal skin.

**Figure 5 genes-13-00797-f005:**
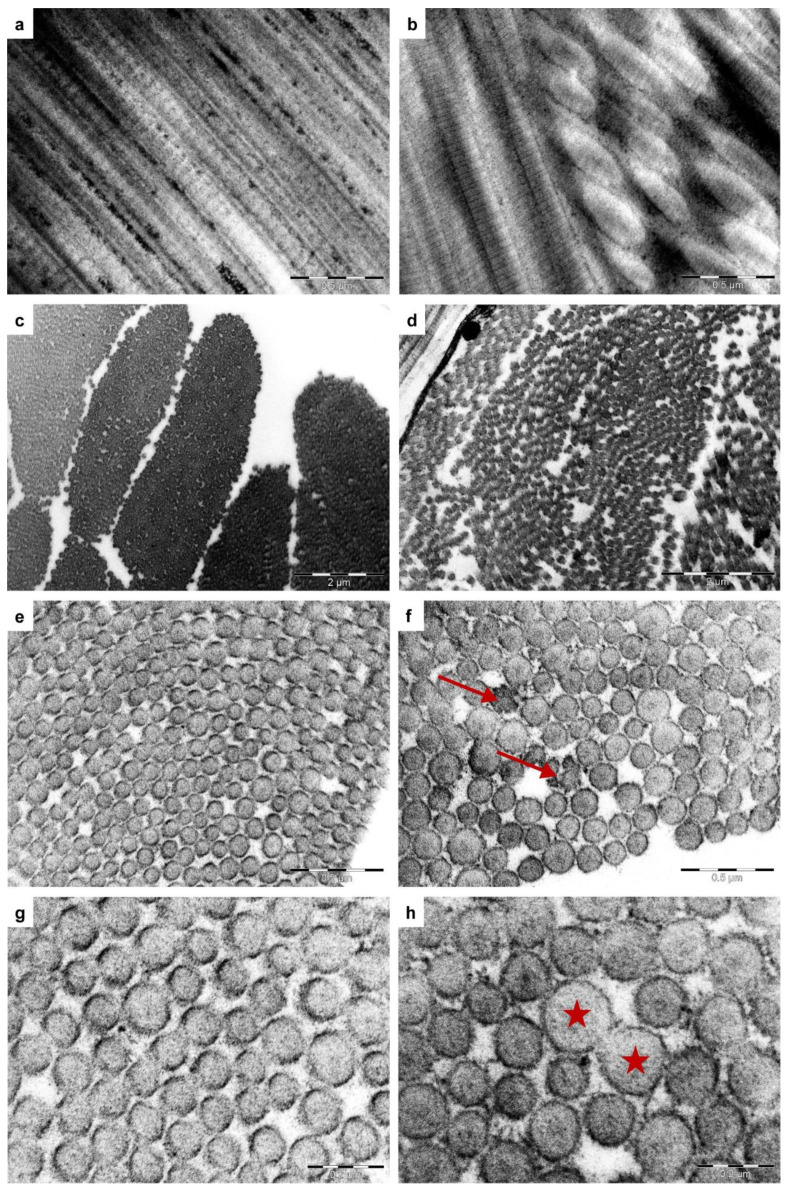
Ultrastructural morphology of the skin from one of the affected cats (case no. 4) and an age-matched control cat. (**a**,**c**,**e**,**g**) Reticular connective tissue of a normal cat’s skin (control cat). (**a**) Longitudinal section of parallel-aligned collagen fibrils (×53.000). (**c**) Collagen fibers are composed of densely packed fibrils (cross-section, ×11.500). (**e**,**g**) Details of cross-sectioned collagen fibers. Consistent diameters of collagen fibrils with regularly shaped and almost round outlines (×53.000, ×110.000). (**b**,**d**,**f**,**h**) Representative collagen fibril abnormalities observed in the reticular dermis of case no. 4 cat. (**b**) Longitudinal sections show disordered, curled fibrils (×53.000). (**d**) Collagen fibers are loosely packed (cross-section, ×11.500). (**f**) Collagen fibrils with irregular outlines (arrows, ×53.000) and variable diameters (**h**). (**h**) Asterisks indicate fibrils with a diameter almost twice as large as surrounding fibrils with a normal diameter (×110.000).

**Figure 6 genes-13-00797-f006:**
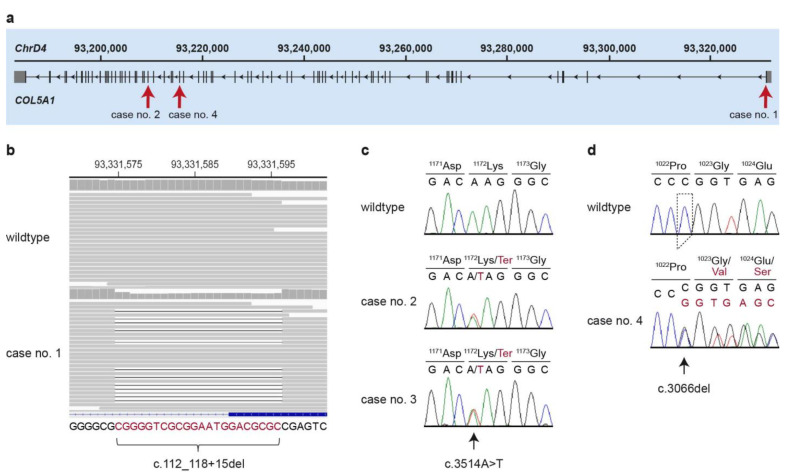
Details of the *COL5A1* variants. (**a**) Overview of the major transcript isoform (XM_023242950.1) of the *COL5A1* gene with the intronic and exonic regions. The positions of all three identified variants are indicated by arrows. (**b**) Integrative Genomics Viewer (IGV) screenshot showing the short-read alignments of a control and the EDS-affected cat (case no. 1) at the position of the deletion. The heterozygous deletion is visible in the short-read alignments and the reduced coverage at the deleted bases. The sequence at the bottom represents the sequence of the coding strand in 3’ to 5’ orientation. The deleted bases are indicated in red. (**c**) Representative electropherograms of a control and two EDS-affected cats (case no. 2 and 3) are shown. The variable position is indicated by an arrow and the amino acid translations are given. Mutant alleles are indicated in red. (**d**) Representative electropherograms of a control and an EDS-affected cat (case no. 4) are shown. The amino acid translations of the wildtype and mutant alleles are indicated.

**Table 1 genes-13-00797-t001:** Results of variant filtering in the affected cats against 54 control genomes.

Filtering Step	Variants Case No. 1		Variants Case No. 2		Variants Case No. 4	
	hom	het	hom	het	hom	het
all variants	5,758,034	9,091,644	4,612,879	8,593,879	4,949,250	8,310,846
private variants	44,007	938,018	6242	194,315	11,606	168,540
protein-changing private variants	91	2901	15	880	51	775
in 20 known EDS candidate genes	0	1	0	1	0	3

**Table 2 genes-13-00797-t002:** Variant designations of the identified *COL5A1* variants according to Human Genome Variation Society (HGVS) nomenclature.

Cats	HGVS Variant Designations		
	Genomic (felCat9.0)	mRNA (XM_023242950.1)	Protein (XP_023098718.1)
case no. 1	ChrD4:93,331,577_93,331,598del	c.112_118+15del	r.spl?
cases no. 2 and 3	ChrD4:93,209,345T>A	c.3514A>T	p.(Lys1172*)
case no. 4	ChrD4:93,215,496del	c.3066del	p.(Gly1023Valfs*50)

## Data Availability

The accessions for the sequence data reported in this study are listed in Appendix A.

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
