# Peer review of "Independent COL5A1 Variants in Cats with Ehlers-Danlos Syndrome"

_genes, 2022, doi:10.3390/genes13050797_

Round 1
Reviewer 1 Report
The manuscript is clear and relevant to identifying causative genetic variants underlying classical EDS in cats. Revealing genetic variation together with clinical symptoms makes the manuscript important.
Author Response
Reviewer 1 did not request any changes. We thank the reviewer for the positive evaluation.
Reviewer 2 Report
Dear authors,
thank you very much for submitting your manuscript to Genes. Your work is outstanding.
I have a few comments/suggestions for you:
(1) I totally agree with you about what you have stated on lines 216 & 217 about case n.3; but, I'm not quite sure if you can include that case into Table 2 since genotyping was not confirmed. As you stated in the DISCUSSION section, you have identified three variants.
(2) DISCUSSION SECTION (the last paragraph, lines 289 to 296): Some non-dermatologist readers can get confused about case n.4 and claim that this case could have itch behavior secondary to EDS (line 285-286, case n.2). On the other hand, Titeux et al. welfare score has not been validated. Perhaps, you want to add more information about how dermatologists consider feline head & neck pruritus as a reaction pattern secondary to allergic dermatitis in most cats. It is just to clarify that case n.4 still were considered allergic in spite of the negative ID test and serology. It's just a thought.
Thank you very much
Author Response
(1)
I totally agree with you about what you have stated on lines 216 & 217 about case n.3; but, I'm not quite sure if you can include that case into Table 2 since genotyping was not confirmed. As you stated in the DISCUSSION section, you have identified three variants.
Response: In lines 216 & 217 we gave the rationale why we sequenced only the genome of case no. 2 (and not its sister, case no. 3). As soon as we had identified the causative variant in case no. 2, we experimentally verified (by Sanger sequencing) that the related case no. 3 carried the same genetic variant. This is described in lines 231-233 of the revised manuscript and documented in figure 6c. We chose this approach to save the costs for sequencing an additional genome.
(2)
DISCUSSION SECTION (the last paragraph, lines 289 to 296): Some non-dermatologist readers can get confused about case n.4 and claim that this case could have itch behavior secondary to EDS (line 285-286, case n.2). On the other hand, Titeux et al. welfare score has not been validated. Perhaps, you want to add more information about how dermatologists consider feline head & neck pruritus as a reaction pattern secondary to allergic dermatitis in most cats. It is just to clarify that case n.4 still were considered allergic in spite of the negative ID test and serology. It's just a thought.
Response: We thank the reviewer for this valuable comment and agree that this might be irritating that despite negative allergy testing the cat still was diagnosed with feline atopic syndrome (FASS).
Diagnosing atopic dermatitis can be difficult since available tests are not diagnostic tests. Tests like IDT or serology only help identify the triggering agent and sometimes can be negative (intrinsic atopic dermatitis). The Favrot’s criteria help to establish the diagnosis of atopic dermatitis - similarly, the criteria described by Titeux et al help to exclude a psychogenic component for the itch. Since Titeux’s scoring system is not validated we used the phrase not suggestive (line 165) instead of excluding in the text.
In our case, the clinical presentation and the fact that the cat showed a seasonal exacerbation as well as the response to histamines (partial) and steroids (complete) suggested an underlying allergic cause. A neurological component of the itch seemed therefore unlikely, but could not be completely excluded, without further diagnostic imaging or necropsy, which was declined by the owner.
We also added language regarding the head & neck reaction pattern in regard to FASS. Please find the amendments in the discussion text (line 292-303).
Reviewer 3 Report
Dear authors, I think your article is interesting to show that genetic diseases are foud in cats as well as in humans.
Although your contribution is good quality, I believe it is necessary to point out some aspects to be clarified in ther text on your part:
1) Variants are not classified. Could you talk about it? On the basis of whic h criteria are they lcassified as pathogenetic?
2) For the intronic variant in position+15, were any predictions made by the tools for prediction of the splicing events?....The use of porgrams is not recommended !
3) You talk about " de novo" events......but have you done a segregation study?
4) It could be useful to consider a bibliographic reference regarding epilepsyand to specify in the text that in humans with this syndrome there are different neurological sympotms as reported in the literature, citing a review of Child's Nerv Syst (2011) 27 : 365-371 "Ehlers-Danlos syndrome and neurological feature:a review
5) Correct some typos found in the text
Author Response
(1)
Variants are not classified. Could you talk about it? On the basis of which criteria are they classified as pathogenetic?
Response: We thank the reviewer for this valuable comment. To the best of our knowledge, consensus criteria for the classification of variants have not yet been established in domestic animals. However, it is relatively straightforward to apply human criteria to cats. We therefore added a paragraph at the end of the discussion explaining our classification of all three reported variants as "pathogenic" according to human ACMG/AMP criteria.
(2)
For the intronic variant in position+15, were any predictions made by the tools for prediction of the splicing events?....The use of programs is not recommended !
Response: The c.112_118+15del variant deletes 7 exonic and 15 intronic bases. We fully agree with the reviewer that it is impossible to reliably predict the consequences on splicing of this variant with an existing computer program. We tried to investigate this experimentally. However, unfortunately, the available skin samples were not of sufficient quality and our attempts at amplifying a cDNA fragment from isolated RNA failed.
(3)
You talk about " de novo" events......but have you done a segregation study?
Response: Yes, for case no. 4, we experimentally demonstrated the de novo mutation event. The data are shown in Figure S1.
(4)
It could be useful to consider a bibliographic reference regarding epilepsy and to specify in the text that in humans with this syndrome there are different neurological symptoms as reported in the literature, citing a review of Child's Nerv Syst (2011) 27 : 365-371 "Ehlers-Danlos syndrome and neurological feature:a review
Response: We added the reference as suggested (ref. 39 in the revised manuscript).
(5)
Correct some typos found in the text
Response: We checked for typographical errors and revised them as best as we could.